# The Impact of the COVID-19 Pandemic on Antimicrobial Resistance and Management of Bloodstream Infections

**DOI:** 10.3390/pathogens12060780

**Published:** 2023-05-30

**Authors:** Vasilios Petrakis, Maria Panopoulou, Petros Rafailidis, Nikolaos Lemonakis, Georgios Lazaridis, Irene Terzi, Dimitrios Papazoglou, Periklis Panagopoulos

**Affiliations:** 1Department of Infectious Diseases, 2nd University Department of Internal Medicine, University General Hospital Alexandroupolis, Democritus University Thrace, 68132 Alexandroupolis, Greece; prafaili@med.duth.gr (P.R.); irene.terzi@gmail.com (I.T.); dpapazog@med.duth.gr (D.P.); ppanago@med.duth.gr (P.P.); 2University Lab of Microbiology, University General Hospital Alexandroupolis, Democritus University Thrace, 68132 Alexandroupolis, Greece; mpanopou@gmail.com (M.P.); nikolaoslemonakis@gmail.com (N.L.); glazarid@icloud.com (G.L.)

**Keywords:** COVID-19 pandemic, antimicrobial resistance, infectious disease consultation, multidrug-resistant bacteria, infection prevention and control group, antibiotic stewardship

## Abstract

Introduction: The pressure of the COVID-19 pandemic on healthcare systems led to limited roles of infectious diseases services, increased rates of irrational use of antimicrobials, and incidence of infections by multidrug-resistant microorganisms. The aim of the present study is to evaluate the incidence of antimicrobial resistance and the management of bloodstream infections before and during the COVID-19 pandemic at the University General Hospital of Alexandroupolis (Greece). Materials and Methods: This is a retrospective study conducted from January 2018 to December 2022. Data were collected from the University Microbiology Laboratory per semester regarding the isolated strains of Gram-positive and -negative bacteria in blood cultures and respiratory samples in hospitalized patients in medical and surgical wards and in the intensive care unit (ICU). Additionally, bloodstream infections with requested infectious disease consultations were reported (n = 400), determining whether these were carried out via telephone contact or at the patient’s bedside. Demographic data, comorbidities, focus of infection, antimicrobial regimen, duration of treatment, length of hospitalization, and clinical outcome were analyzed. Results: A total of 4569 strains of Gram-positive and -negative bacteria were isolated. An increasing trend was reported compared to the pre-pandemic period in the incidence of resistant Gram-negative bacteria, particularly in ICUs. Prior antimicrobial use and the rate of hospital-acquired infections were increased significantly during the pandemic. In the pre-pandemic period 2018–2019, a total of 246 infectious disease consultations were carried out, while during the period 2020–2022, the number was 154, with the percentage of telephone consultations 15% and 76%, respectively. Detection of the source of infection and timely administration of appropriate antimicrobial agents were more frequently recorded before the pandemic, and 28-day mortality was significantly reduced in cases with bedside consultations. Conclusion: The empowering of infectious disease surveillance programs and committees, rational use of antimicrobials agents, and bedside infectious disease consultations are vital in order to reduce the impact of infections caused by multidrug-resistant strains.

## 1. Introduction

The COVID-19 pandemic caused by severe acute respiratory syndrome coronavirus 2 (SARS-CoV-2) was spread rapidly and overwhelmed healthcare systems worldwide [1]. Simultaneously, the rise in multidrug-resistant (MDR) infections continues to threaten public heath leading to high rates of morbidity, mortality, and economic loss [1]. Antimicrobial resistance (AMR) is estimated to lead to 700,000 deaths globally each year [2]. AMR can be defined as an increasing resistance to antibiotics that undermines the ability to treat common and serious infectious diseases [2]. The estimated number of deaths due to infections with multiple drug-resistant pathogens by the year 2050 is expected to reach 10 million/year in the case of not applying a strict action plan to combat AMR [2]. The WHO declared AMR as one of the top 10 global health threats in 2015 and, although often more silent than the COVID-19 pandemic, it can have similar devastating consequences [3]. A European Centre for Disease Control and Prevention (ECDC) study in 2015 on the health burden of antimicrobial resistance measured in numbers of cases, attributable deaths, and disability-adjusted life-years (DALYs) concluded that 33,110 deaths and 874,541 DALYs were caused by infections with antibiotic-resistant bacteria which were mainly healthcare-associated (75%) [4]. Published data from the ECDC in November 2022 underlined that more than 35,000 people die from antimicrobial-resistant infections in the EU/EEA each year, and between 2012 and 2021 the consumption of ‘broad-spectrum’ antibiotics in hospitals was increased by 15% and the proportion of ‘reserve’ antibiotics more than doubled [5]. 

The increase in antimicrobial resistance is a potential consequence of the COVID-19 pandemic. Several recent reports have described an increase in multidrug-resistant bacteria during the COVID-19 pandemic [6,7]. The cause is multifactorial, but a major reason is the high rate of antimicrobial agent utilization in COVID-19 patients despite the relatively low rates of co- or secondary infections [6]. A retrospective study found that the incidence of carbapenem-resistant *Enterobacterales* colonization in ICU patients increased from 6.7% in 2019 to 50% in March–April 2020 [7]. Because of the COVID-19 emergency in healthcare systems, planned activities were deprioritized and already implemented preventive measures were reversed [8]. The pandemic has put tremendous strain on healthcare systems, diverting resources, personnel, and attention away from AMR diagnosis and management while AMR studies were hampered and surveillance programs were de-emphasized or stopped [9]. 

The aim of the present study is to evaluate the impact of the COVID-19 pandemic on antimicrobial resistance by estimating the nonsusceptibility rates of isolates in blood cultures and respiratory samples and on the management of bloodstream infections in University General Hospital of Alexandroupolis (Greece) during the period 2018–2022. Surveillance programs and reports of AMR before and after the COVID-19 pandemic are vital in order to better understand the impact of pandemic and decide appropriate interventions.

## 2. Materials and Methods

This was a retrospective study conducted in the University Lab of Microbiology and Department of Infectious Diseases of the University General Hospital of Alexandroupolis (Greece). Data from lab records and routine care patient charts during the period 1 January 2018 to 31 December 2022 were retrospectively analyzed. The study was carried out in accordance with the Helsinki Declaration of Human Rights.

During the 5-year period, routine susceptibility data of Gram-negative and Gram-positive bacterial isolates from blood and respiratory specimens of hospitalized patients in surgical and medical wards and intensive care units (ICU) were reported emphasizing to the most clinically important species (*Acinetobacter baumannii*, *Klebsiella pneumoniae*, *Pseudomonas aeruginosa*, *Enterococcus faecium*, and *Staphylococcus aureus*). From each patient, only the first isolate of a given species recovered was included, regardless of susceptibility profile or specimen type. The classification of the isolates as susceptible, intermediate, or resistant was based on the Clinical and Laboratory Standards Institute (CLSI) and the European Committee on Antimicrobial Susceptibility Testing (EUCAST) guidelines [10,11]. The EUCAST breakpoints, valid since January 2019, which established the susceptibility cut-off points based on antibiotic dose and mode of administration, were adopted by our lab in January 2023 after the end of study period. The antimicrobial susceptibility testing was performed by an automated antimicrobial susceptibility testing system for the minimal inhibitory concentration (MIC) determination. During the period 2018–2019, the University General Hospital of Alexandroupolis contained 650 beds in wards and 15 in the ICU. The management of the COVID-19 pandemic led to an increased number of beds: 700 in the wards and 21 in the ICU. The hospital occupancy remained similar between the pre-pandemic period and during the COVID-19 pandemic (72% vs. 74%), while scheduled surgeries were postponed and non-COVID-19 medical hospitalizations were limited during the first year of the pandemic. 

The number of non-susceptible isolates was divided by the number of isolates tested in order to determine the non-susceptibility rate for every assessment period, defined as a semester. The isolates were analyzed based on microorganism, ward type (ward, ICU), specimen type, and antibiotic. The administration of antibiotics prior to pathogen isolation was reported. The percentage of isolates attributed to healthcare-associated infections was estimated for each year. The clinical outcome was evaluated by documenting the risk of death and increased hospital stay due to the isolated pathogens. Multivariable analysis was performed for the attributable risk of death and prolonged hospital stay and the difference between patients with infections by multidrug-resistant bacteria and patients infected by susceptible bacteria was estimated. 

Additionally, the cases of bloodstream infections with reported consultation by an infectious diseases specialist during the period 2018–2022 were documented. The results of blood cultures from wards and ICU were routinely reviewed by the infectious diseases service in order to identify resistant isolates and design the appropriate management. Infectious disease consultations were conducted after the request of the primary clinical team. The patients were divided into two groups based on period of time, group A for the pre-pandemic period (2018–2019) and group B for the COVID-19 pandemic period (2020–2022). Type of consultation (bedside or via telephone), demographic characteristics of patients (age, gender), comorbidities using the Charlson comorbidity index (CCI), duration of symptoms before consultation, type of infection (community or hospital acquired), foci of infection, clinical outcome (mortality within 28 and 90 days after first pathogen isolation), and features of antibiotic therapy (duration, combination of antibiotics) were analyzed. A community-acquired infection was defined as an infection contracted outside of a healthcare facility or an infection present at the time of admission [11]. The term healthcare-associated infections (HCAIs) was used for infections occurred while receiving healthcare or developed in a hospital or other healthcare facility firstly appeared 48 h or more after hospital admission or within 30 days after having received healthcare [12]. HCAIs include central line-associated bloodstream infections, catheter-associated urinary tract infections, and ventilator-associated pneumonia [12]. 

Quality indicators for the present study included: (a) identification of focus of infection, (b) performing repeat blood cultures at 48 to 96 h, (c) treatment of uncomplicated infections with 14 days of iv antibiotics, (d) treatment of complicated infections with a minimum of 28 days of iv antibiotics, and (e) bacteremia recorded in the hospital discharge summary. The outcome measures were assessed by the following factors: (a) defervescence within 7 days, (b) duration of hospital stay, (c) death at 28 and 90 days after first positive blood culture, and (d) recurrent disease. Death was attributed to bloodstream infection if the blood culture was positive at the time of death and the associated symptoms and signs were still present. 

Statistical analysis of the data was performed using IBM Statistical Package for the Social Sciences (SPSS), version 19.0 (IBM Corp., Armonk, NY, USA). The normality of quantitative variables was tested with Kolmogorov–Smirnov test. Normally distributed quantitative variables are expressed as the mean ± standard deviation (SD), while non-normally distributed quantitative variables are expressed as the median value and range. Qualitative variables were expressed as absolute and relative (%) frequencies. The Fisher exact test was used to compare categorical variables and the Mann–Whitney test to compare continuous variables. Time-to-event outcomes were analyzed by the Cox proportional hazard model. A multivariable Cox proportional hazard model, adjusted for age, sex, hospital-acquired infection, and CCI, was used to compare the 28- and 90-day mortality between the two groups. Student’s *t*-test, Mann–Whitney U-test, and chi-square test were used to determine differences in demographic and clinical characteristics between the two groups of patients. All tests were two-tailed and statistical significance was considered for *p* values < 0.05.

## 3. Results

During the period 2018–2022 (pre-pandemic period group A, 2018–2019, and pandemic period group B, 2020–2022), a total number of 4569 bacteria were isolated in blood and respiratory samples from patients hospitalized in wards (group A, n = 1285 and group B, n = 2068) and ICUs (group A, n = 393 and group B, n = 823). Number of isolates by semester is shown for ICU in Figure 1 and for wards in Figure 2. After the initiation of the COVID-19 pandemic, a significant increase is reported in numbers of isolated bacteria and eventually for Gram-negative bacilli and *Enterococcus* isolates.

In *Acinetobacter baumannii* isolates from hospitalized patients in ICUs, the non-susceptibility to carbapenems remained high during the whole study period (Figure 3). The percentage of *Acinetobacter baumannii* isolates from blood and respiratory samples in ICUs resistant to meropenem was 92.6% in the first semester of 2018 and 97.9% in the second semester of 2022 (*p* < 0.001). The non-susceptibility to meropenem in wards increased from 82.3% in the first semester of 2018 to 91.6% in the second semester of 2022 (*p* < 0.001). A significant difference was found in the non-susceptibility trend for amikacin, ranging from 81.6% in the first semester of 2018 to 93.4% in the second semester of 2022 in ICUs and from 63.2% to 74.5% respectively in wards (*p* < 0.001). A decreasing trend was reported in the slope of the non-susceptibility for colistin in ICU samples during the pre-pandemic period (from 43.6% to 41.4%, *p* = 0.05) followed by an increasing trend during the pandemic period (from 42.5% to 59.6%, *p* < 0.001).

In *Klebsiella pneumonia* isolates from blood and respiratory samples in ICUs, a decreasing trend in non-susceptibility to meropenem and colistin prior to COVID-19 pandemic initiation (Figure 4). However, during pandemic period the rate of non-susceptibility to colistin was increased from 58.3% in the second semester of 2019 to 71.8% in the second semester of 2022 and to meropenem from 79.8% to 92.4%, respectively (*p* < 0.001) (Figure 4). Similar findings were documented in the wards with 56.4% of *Klebsiella pneumonia* isolates resistant to colistin and 72.6% to meropenem during the second semester of 2022 (Figure 4).

In *Pseudomonas aeruginosa* isolates from patients hospitalized in wards and ICUs, statistically significant changes were found in the rates of non-susceptibility trends before and during the COVID-19 pandemic (Figure 5). The non-susceptibility of isolates in ICUs to meropenem was increased from 43.5% in the first semester of 2018 to 53.6% in the second semester of 2022 (*p* < 0.001). Similarly, the non-susceptibility to amikacin was reported 41.3% in the first semester of 2018 and was increased to 53.6% in the pandemic period (*p* < 0.001). In wards the non-susceptibility to levofloxacin remained high during 2018–2022, ranging from 42.5% in the first semester of 2018 to 49.6% in the second semester of 2022 (*p* < 0.001). Despite the decreasing trend of non-susceptibility to meropenem and amikacin in wards in the first semester of 2020, during the pandemic period the rate of non-susceptibility to meropenem increased to 42.5% and to amikacin to 38.6% (*p* < 0.001).

A decreasing rate of non-susceptibility to vancomycin was reported in the second semester of 2019 for *Enterococcus feacium* isolates in wards. However, during the pandemic period, the non-susceptibility rate to vancomycin increased from 33.5% to 44.6% in the second semester of 2022 (*p* < 0.001) (Figure 6).

In *Staphylococcus aureus* isolates in wards during the pre-pandemic period, a decreasing trend of non-susceptibility to methicillin was found, from 33.2% in the first semester of 2018 to 30.5% in the first semester of 2019 (*p* = 0.052). During the COVID-19 pandemic, the rate was increased significantly to 38.6% (*p* < 0.001) (Figure 7).

The estimated percentage of isolates attributed to hospital acquired infections was increased from 42% in 2018 to 60% in 2021 and 64% in 2022 (*p* = 0.05). Antibiotic consumption before the isolation of bacteria in blood and respiratory samples was reported in high rates during 2018–2022, ranging from 53% in 2018 to 78% in 2022 (*p* = 0.005). The results of multivariable logistic regression analysis are shown in Table 1. The increased numbers of isolates in wards and ICU and high rates of non-susceptibility to antibiotics led to increased risk of death and prolonged hospital stay, particularly for Gram-negative bacilli. 

A total of 400 adults with bloodstream infections and reported infectious disease consultations were eligible for inclusion in the study. During the pre-pandemic period (group A), 246 consultations were documented, 85% carried out at bedside. During the COVID-19 pandemic (group B), the number of consultations was lower (n = 154) and mainly conducted via telephone (76%). Demographic data (age, gender) of patients between the two groups were similar (Table 2). The first consultation was reported within 24 h from symptom onset in 64.2% of patients in group A while 47.4% in group B (Table 2). The proportion of patients with Charlson comorbidity index Score ≥ 3 was approximately 40% for both groups (Table 3). The proportion of patients with prosthetic material was higher in group B compared to group A (30.5% vs. 22.8%, *p* < 0.02).

Overall, approximately 60% of bloodstream infections were hospital acquired and 30% were community acquired (Table 4). The frequency of hospital-acquired infections and multidrug-resistant isolated bacteria were higher in group B (63.6% vs. 61%, *p* < 0.001 and 37% vs. 33.7%, *p* < 0.001, respectively). The most common foci of infection were skin and soft tissue infections, central venous catheters associated infections, respiratory infections and osteomyelitis (Table 4). The proportion of patients with an unknown focus was higher in group B (11.7% vs. 6.5%, *p* < 0.04). Certain foci of infection such as thrombophlebitis, implanted vascular device, prosthetic joint infections, urinary tract infections, and intra-abdominal infections were more frequently identified in group A compared to group B (Table 4). The frequency of a complicated infection was similar between the two groups (54.5% vs. 56.5%, *p* < 0.001).

The percentage of patients for whom bloodstream infection was complicated with septic shock was 3.3% in group A and 4.5% in group B (*p* = 0.118) (Table 5). Need for hospitalization in intensive care units (4.5% vs. 7.8%, *p* = 0.245) and length of hospital stay (29 days vs. 30 days, *p* = 0.457) were not significantly different between the two groups (Table 5). The duration of antibiotic treatment was longer in group A compared to group B (15 days vs. 11 days, *p* = 0.04). Higher frequency of treatment with combination of antibiotics was reported in group A (10.6% vs. 7.1%, *p* < 0.001). The mortality rate within 28 days and 90 days was significantly higher in group B, with mainly telephone infectious disease consultations (Table 5). The proportion of cases in which new blood cultures after treatment initiation were collected and repeated consultations were carried out was higher in group A with bedside infectious disease consultations. There was no significant difference in recurrent bacteremia between the two groups.

## 4. Discussion

The COVID-19 pandemic induced a new burden on health systems worldwide and aggravated existing health challenges in many aspects of global health [13]. According to the annual Tripartite AMR Country Self-Assessment Survey 2020–21, 151 (94%) of 161 countries ascribed the pandemic as having impacted their national response to control the problem of antimicrobial resistance [14]. An acute increase in the burden of antimicrobial resistance was a feared outcome of the pandemic, but this speculation had not been comprehensively measured [13,14]. The present study showed that the rates of multidrug-resistant bacteria during the COVID-19 pandemic were increased significantly. An increasing trend was reported in numbers of isolated Gram-negative bacteria, especially for *Acinetobacter baumannii* non-susceptible to colistin and *Pseudomonas aeruginosa* non-susceptible to carbapenems and quinolones, leading to increased risk of death and longer hospital stay. The management of bloodstream infections was altered during the COVID-19 pandemic based on the results of our study, while infectious disease consultations were mainly conducted via telephone and not bedside. Telephone infectious disease consultations were associated with higher rates of mortality and lower probability for appropriate antimicrobial scheme and repeated clinical evaluation. 

A special report by the Center for Disease Control and Prevention (CDC) with country-level estimates of the effect of COVID-19 on antimicrobial resistance in the USA highlights a devastating undoing of progress in efforts to control antimicrobial resistance [15]. Although deaths from antimicrobial resistance were reduced by 18% from 2012 to 2017, including a 30% reduction in US hospitals, during 2020 a 15% increase in drug-resistant nosocomial infection rates was reported compared with the previous year [15]. Pathogen–drug combinations, classified as critical by WHO based on their risk to human health, showed alarming increases in rates of infection since 2019, especially in rates of carbapenem-resistant *Enterobacterales* infections up 35%, and carbapenem-resistant *Acinetobacter* increasing by 78% [15]. Similar findings were documented in the results of the Greek Electronic System for the Surveillance of Antimicrobial Resistance (WHONET-Greece) which analyzed routine susceptibility data of 17,837 Gram-negative and Gram-positive bacterial isolates from blood and respiratory specimens of hospitalized patients in nine COVID-19 tertiary hospitals in two periods, January 2018–March 2020 and April 2020–March 2021 [1]. Increases were observed in the number of bloodstream and respiratory isolates from ICU patients in the last 6 months of the study period (October 2020–March 2021) mainly due to *A. baumannii* isolates in both blood and respiratory specimens and *E. faecium* blood isolates compared to the previous 6 months [1]. Significant differences were found in the slope of non-susceptibility trends of *Acinetobacter baumannii* blood and respiratory isolates to amikacin, tigecycline, and colistin and of *Pseudomonas aeruginosa* respiratory isolates to imipenem, meropenem, and levofloxacin [1]. However, decreasing nonsusceptibility trends in respiratory P. aeruginosa isolates were reported in the results of WHONET [1]. In our study the nonsusceptibility rates of P. aeruginosa isolates were significantly increased. 

Few studies try to approach the causes of the significant impact of the COVID-19 pandemic on antimicrobial resistance. The antibiotic consumption was irrational during the pandemic, while the rates of microbiologically confirmed bacterial coinfection were low [16,17,18,19,20]. In a meta-analysis including 24 studies and focusing on bacterial co-infections in patients hospitalized for COVID-19, co-infection was reported in 3.5% (95%CI: 0.4–6.7%) and secondary infection in 14.3% (95%CI: 9.6–18.9%) of patients with COVID-19 [21]. The reported bacterial infection was 6.9%, ranging from 5.9% in hospitalized patients to 8.1% in critically ill patients [21]. Many microorganisms have been reported as co-pathogens, including *Streptococcus pneumoniae*, *S. aureus*, *K. pneumoniae*, *Mycoplasma pneumoniae*, *Chlamydophila pneumoniae*, *Legionella pneumophila*, *E. coli*, *P. aeruginosa*, *S. maltophilia*, *A. baumannii*, *Mycobacterium tuberculosis*, *Candida* spp., *Aspergillus* spp., and viruses such as influenza, rhinovirus/enterovirus, parainfluenza virus, metapneumovirus, and human immunodeficiency virus (HIV) [22,23,24,25,26]. The majority of multidrug-resistant microorganisms was developed in patients with severe or critical COVID-19, resulting in prolonged hospitalization and increased mortality rates [22]. 

In our study a high frequency of *A. baumannii* blood and respiratory isolates was observed with high levels of non-susceptibility to carbapenem and colistin throughout the study period. *A. baumannii* is one of the ESKAPE organisms (*Enterococcus faecium*, *Staphylococcus aureus*, *Klebsiella pneumoniae*, *A. baumannii*, *Pseudomonas aeruginosa*, and *Enterobacter* spp.) and remains a therapeutic challenge with constantly increasing resistance [27]. Multidrug-resistant *A. baumannii* isolates have been identified in COVID-19 patients in both blood and respiratory isolates, mainly from ICUs, and with high rates of resistance in almost all widely used antibiotics, such as carbapenems, colistin and tigecycline [28,29,30,31]. Increasing trend was also reported in non-susceptible strains of *Klebsiella pneumoniae* during the COVID-19 pandemic period. In Greece, high rates of carbapenem-resistant *K. pneumoniae* isolates due to carbapenemase-producing strains have been documented since 2002 [32]. A literature review of carbapenem-resistant *Klebsiella pneumoniae* infections in patients hospitalized due to COVID-19 the prevalence of coinfection ranged from 0.35% to 53% [33]. 

Studies have shown that over 70% of patients with COVID-19 receive antibiotics, with the majority constituting broad-spectrum agents such as fluoroquinolones and third-generation cephalosporins [21]. A study including 138 hospitalized patients showed that moxifloxacin, ceftriaxone, and azithromycin were prescribed in 89 (64.5%), 34 (24.6%), and 25 (18.1%) patients, respectively [18]. In a large-scale study with 1099 patients, 58% received intravenous antibiotics [34]. In a smaller Brazilian cohort of 72 hospitalized patients, 84.7% had received intravenous antibiotic therapy [35]. Among antibiotics, β-lactams were the main antibiotic category administered during the pandemic period [6,36]. Azithromycin, the macrolide most used alone or in combination with β-lactams, vancomycin, carbapenems, tigecycline, ceftriaxone, and linezolid, which are all classified as critically important antimicrobials (CIA) by WHO, are being widely prescribed during this pandemic [37]. 

Our study findings support that the use of routine bedside consultation in the pre-pandemic period for the management of bloodstream infections is superior to telephone consultation, which was mainly applied during the COVID-19 pandemic. Patients in group A with bedside consultations were more likely to have an identified focus of infection, a better clinical outcome, and receive longer courses of antimicrobial therapy often with combinations of antibiotics. Longer treatment courses in group A were probably due to higher rates of inpatient follow-up estimation by the infectious disease specialist resulting in improved compliance to recommendations, higher frequency for repeated blood cultures, and appropriate evaluation of clinical and lab biomarkers. An observational cohort study of 571 adults with *Staphulococcus aureus* bacteraemia at a teaching hospital in the United Kingdom between July 2006 and December 2012 showed that bedside consultation was associated with lower mortality at 30 days compared to telephone consultation (12% vs. 22%, *p* < 0.07) [38]. Another study in Helsinki University Central Hospital in Finland including 342 adults with at least 1 positive blood culture for S. aureus were retrospectively analyzed [39]; 72% of patients received bedside consultations, 18%received telephone consultations, and 10% received no consultation [39]. Patients with bedside consultation had lower mortality than patients with telephone consultation at 7 days (OR, 0.09; 95% CI, 0.02–0.49; *p* = 0.001; 1% vs. 8%), at 28 days (OR, 0.27; 95% CI, 0.11–0.65; *p* = 0.002; 5% vs. 16%), and at 90 days (OR, 0.25; 95% CI, 0.13–0.51; *p* < 0.0001; 9% vs. 29%) [39]. The above findings suggest that bedside consultations are superior to telephone consultations and should become the standard of care for the management of bloodstream infections. 

One of the limitations of the study is its retrospective nature. Further studies in larger patient series with simultaneous analyze of data from other hospitals are needed in order to better determine the impact of the COVID-19 pandemic on antimicrobial resistance. The study was not designed as a controlled study and thus the contribution of uncontrolled variables such as patient population, length of hospital stay, and infection duration to the increase in the number of resistant isolates was not widely analyzed. Another limitation of the study is the low numbers of isolates, particularly in ICUs. However, the documented high rates of resistant isolates in our study are similar to published data during the COVID-19 pandemic and underline that antimicrobial resistance is a global health challenge. 

## 5. Conclusions

Our study provides significant results for the possible impact of the COVID-19 pandemic on antimicrobial resistance, which could be valuable for possible effective interventions. The high prevalence of multidrug-resistant pathogens, mainly due to carbapenem-resistant Gram-negative bacilli, is a major public health problem. The pressure of the COVID-19 pandemic on healthcare systems was enormous, limiting the surveillance programs of infectious diseases. Antimicrobial resistance is an emerging silent parallel pandemic. The present study showed an increased frequency of isolated multidrug-resistant bacteria in blood and respiratory samples during the COVID-19 period, with high non-susceptibility rates to antibiotics. The high rates of multidrug-resistant bacteria increased significantly the risk of death and prolonged hospital stay due to complications. The percentage of hospital-acquired infections and the consumption of antibiotics were also increased during the COVID-19 pandemic, amplifying the dimensions of antimicrobial resistance. The management of bloodstream infections was also affected, while infectious disease consultations were conducted mainly via telephone and not at bedside. However, telephone consultations were associated with poorer clinical estimation and outcomes with higher mortality rates. Our study, similar to published data in the literature, concludes that they cannot replace bedside clinical estimation. Empowering of infectious disease surveillance programs and committees and bedside infectious disease consultations are vital in order to reduce the irrational use of antibiotics and the impact of infections by multidrug-resistant microorganisms.

## Figures and Tables

**Figure 1 pathogens-12-00780-f001:**
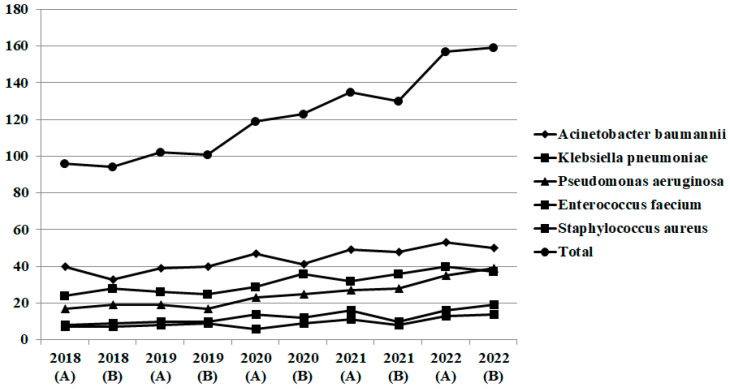
Gram-negative and -positive bacteria isolated in blood and respiratory specimens in intensive care units, ICUs, n = 1216 (group A, pre-pandemic period, n = 393 and group B, pandemic period, n = 823) (2018–2019).

**Figure 2 pathogens-12-00780-f002:**
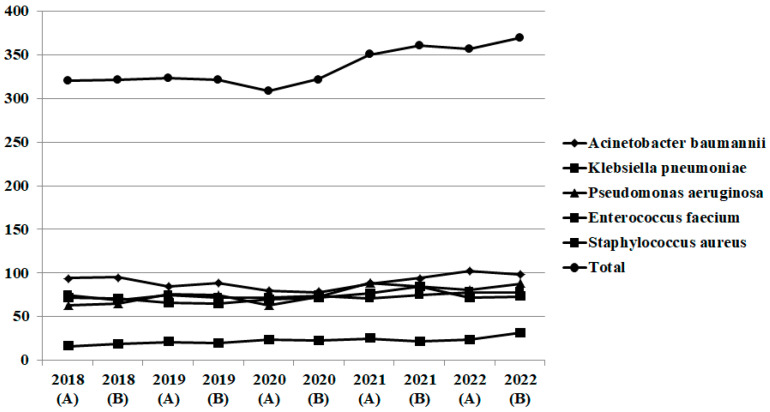
Gram-negative and -positive bacteria isolated in blood and respiratory specimens in surgical and medical wards, n = 3353 (group A, n = 1285 and group B, n = 2068) (2018–2019).

**Figure 3 pathogens-12-00780-f003:**
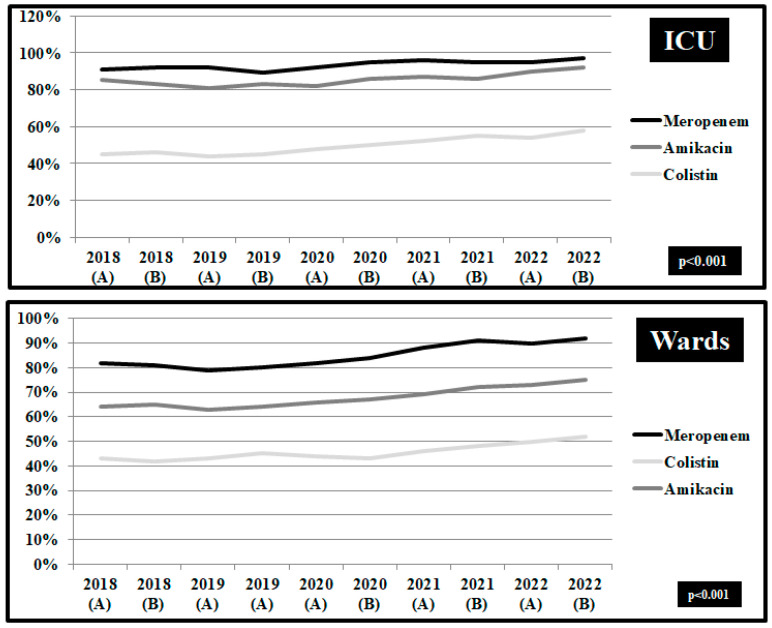
Rates (%) of non-susceptible *Acinetobacter baumannii* isolates from blood and respiratory specimens to meropenem, amikacin, and colistin, per semester, from patients hospitalized in surgical and medical wards and intensive care units, ICUs, 2018–2022. ICUs: pre-pandemic period, group A n = 152 and pandemic period, group B, n = 286. Wards: pre-pandemic period, group A n = 363 and pandemic period, group B, n = 540.

**Figure 4 pathogens-12-00780-f004:**
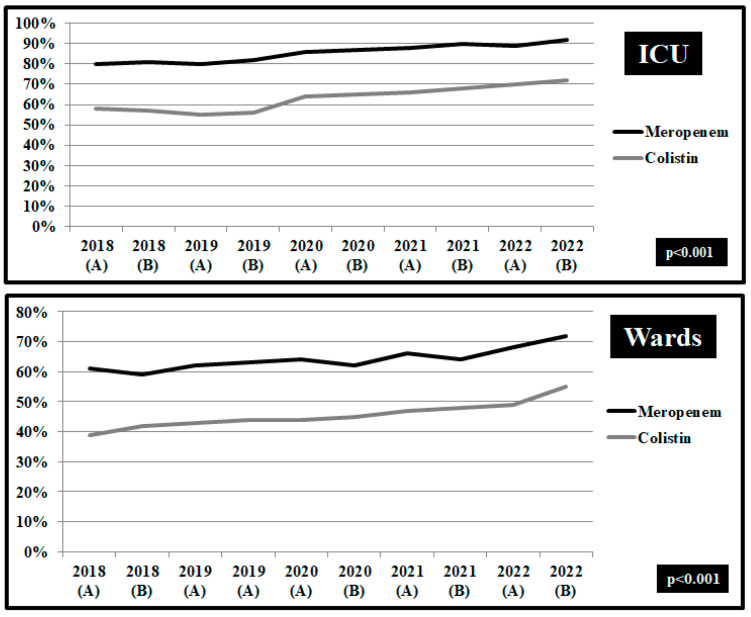
Rates (%) of non-susceptible *Klebsiella pneumoniae* isolates from blood and respiratory specimens to meropenem and colistin, per semester, from patients hospitalized in surgical and medical wards and intensive care units, ICUs, 2018–2022. ICUs: pre-pandemic period, group A n = 103 and pandemic period, group B, n = 210. Wards: pre-pandemic period, group A, n = 291 and pandemic period, group B, n = 448.

**Figure 5 pathogens-12-00780-f005:**
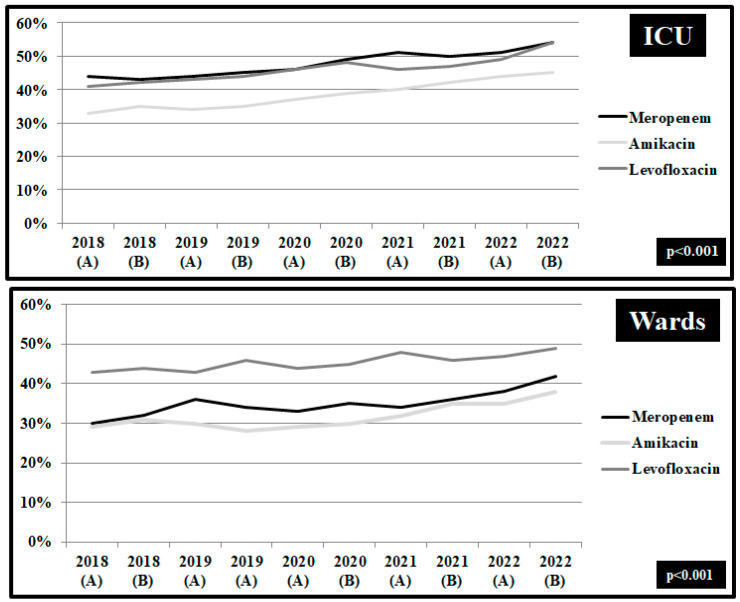
Rates (%) of non-susceptible *Pseudomonas aeruginosa* isolates from blood and respiratory specimens to meropenem, amikacin and levofloxacin, per semester, from patients hospitalized in surgical and medical wards and intensive care units, ICUs, 2018–2022. ICUs: pre-pandemic period, group A n = 72 and pandemic period, group B, n = 177. Wards: pre-pandemic period, group A n = 363 and pandemic period, group B, n = 279.

**Figure 6 pathogens-12-00780-f006:**
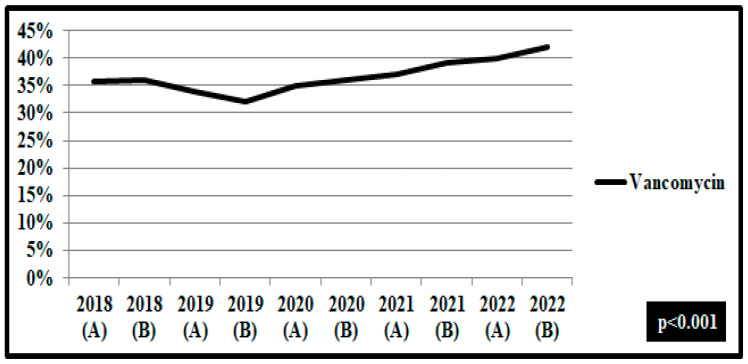
Rates (%) of non-susceptible *Enterococcus feacium* isolates from blood and respiratory specimens to vancomycin, per semester, from patients hospitalized in wards, 2018–2022. ICUs: pre-pandemic period, group A n = 31 and pandemic period, group B, n = 61. Wards: pre-pandemic period, group A n = 76 and pandemic period, group B, n = 150.

**Figure 7 pathogens-12-00780-f007:**
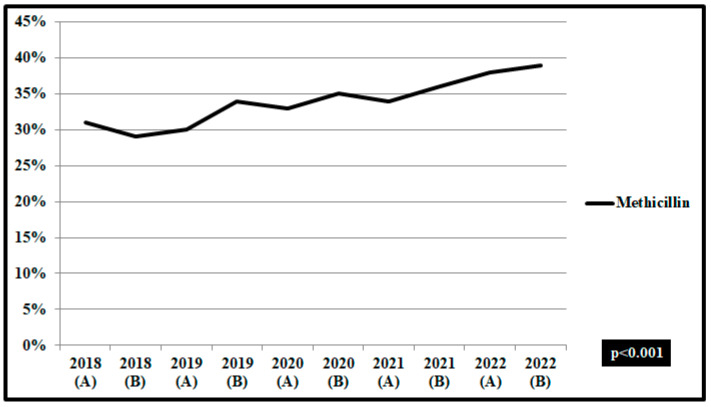
Rates (%) of non-susceptible *Staphylococcus aureus* isolates from blood and respiratory specimens to methicillin, per semester, from patients hospitalized in surgical and medical wards and intensive care units, ICUs, 2018–2022. ICUs: pre-pandemic period, group A n = 37 and pandemic period, group B, n = 87. Wards: pre-pandemic period, group A n = 273 and pandemic period, group B, n = 449.

**Table 1 pathogens-12-00780-t001:** Multivariable logistic regression analysis. Risk of death and prolonged hospital stay in hospitalized patients in wards with isolated Gram-negative/positive bacteria, per year, 2018–2022.

	Risk of Death, OR (95% CI)					Risk of Prolonged Hospital Stay, OR (95% CI)				
	2018	2019	2020	2021	2022	2018	2019	2020	2021	2022
*Acinetobacter baumannii* Carvapenem resistant	3.6 (2.1–4.5) n = 145	3.5 (2.4–5.1) n = 136	3.9 (3.1–6.1) n = 125	4.1 (3.4–6.1) n = 162	4.2 (3.5–5.2) n = 174	7.6 (5.3–9.5) n = 145	7.8 (6.2–9.2) n = 136	7.9 (7.2–9.4) n = 125	7.8 (6.9–9.2) n = 162	8.2 (7.6–10.1) n = 174
*Acinetobacter baumannii* Non Carvapenem resistant	2.1 (1.4–2.8) n = 44	2.2 (1.5–2.7) n = 38	2.4 (1.8–3.1) n = 33	2.3 (1.7–2.9) n = 20	2.4 (1.6–2.9) n = 26	4.3 (3.9–5.4) n = 44	4.5 (3.8–5.9) n = 38	4.3 (3.7–5.4) n = 33	4.4 (3.9–5.8) n = 20	4.5 (3.7–5.3) n = 26
*Klebsiella pneumoniae* Carvapenem resistant	3.1 (2.4–5.1) n = 98	3.3 (2.2–5.1) n = 109	3.6 (2.8–5.7) n = 106	3.5 (2.9–4.3) n = 110	3.8 (2.6–4.1) n = 126	6.5 (5.4–7.9) n = 98	6.7 (5.6–8.4) n = 109	6.7 (6.2–8.6) n = 106	6.9 (5.9–7.8) n = 110	7.1 (6.5–8.9) n = 126
*Klebsiella pneumoniae* Non Carvapenem resistant	2.2 (1.4–2.8) n = 46	1.9 (1.3–2.7) n = 38	2.1 (1.6–2.9) n = 40	2.2 (1.5–3.1) n = 36	2.3 (1.6–2.8) n = 30	3.1 (2.7–4.2) n = 46	3.3 (2.7–4.5) n = 38	3.4 (2.8–4.7) n = 40	3.5 (2.7–4.9) n = 36	3.5 (2.8–4.8) n = 30
*Pseudomonas aeruginosa* MDR, Multidrug resistant	3.1 (2.4–4.7) n = 87	3.4 (2.4–4.9) n = 121	3.8 (2.4–5.9) n = 101	3.7 (2.9– 2.8) n = 131	3.9 (3.1–4.3) n = 129	5.6 (4.6–7.3) n = 87	5.8 (4.5–7.9) n = 121	5.9 (4.5–7.2) n = 101	6.4 (5.9–7.3) n = 131	6.5 (4.9–8.2) n = 129
*Pseudomonas aeruginosa* Non MDR, Non Multidrug resistant	1.9 (1.3–3.7) n = 31	2.0 (1.4–3.8) n = 30	2.1 (1.5–3.7) n = 35	2.1 (1.6–3.9) n = 43	2.2 (1.7–3.8) n = 40	3.2 (2.8–4.3) n = 31	3.3 (2.7–4.9) n = 30	2.9 (2.6–4.5) n = 35	3.0 (2.4–4.2) n = 43	3.1 (2.6–4.9) n = 40
*Enterococcus faecium* VRE, Vancomycin resistant	1.4 (0.9–1.8) n = 8	1.6 (1.1–3.6) n = 12	1.6 (1.2–3.4) n = 15	1.9 (1.1– 2.2) n = 17	2.1 (1.7–2.9) n = 18	3.9 (3.1–5.3) n = 8	3.8 (3.1–5.7) n = 12	4.2 (3.6–6.3) n = 15	4.6 (4.5–7.6) n = 17	4.8 (4.1–7.3) n = 18
*Enterococcus faecium* Non VRE, Non Vancomycin resistant	1.1 (0.7–1.7) n = 27	1.2 (0.7–2.1) n = 29	1.3 (0.8–2.4) n = 32	1.4 (0.9–2.6) n = 30	1.4 (0.9–2.5) n = 36	2.5 (1.9–3.8) n = 27	2.6 (2.1–3.9) n = 29	2.5 (1.8–3.7) n = 32	2.7 (2.0–3.9) n = 30	2.6 (1.9–3.7) n = 36
*Staphylococcus aureus* MRSA, Methicillin resistant	3.0 (2.1–4.9) n = 48	3.0 (2.2–4.7) n = 41	3.1 (2.3– 4.5) n = 52	3.2 (2.4–4.1) n = 59	3.4 (2.8–4.9) n = 55	2.3 (2.1–4.6) n = 48	2.3 (2.1–4.3) n = 41	2.4 (2.1–4.1) n = 52	2.4 (1.8–4.3) n = 59	2.6 (2.2–5.1) n = 55
*Staphylococcus aureus* Non MRSA, Non Methicillin resistant	1.7 (0.8–2.6) n = 95	1.6 (0.9–2.6) n = 90	1.6 (1.0–2.6) n = 90	1.8 (1.1–2.9) n = 98	1.8 (1.0–2.8) n = 90	2.1 (1.4–3.5) n = 95	2.2 (1.4–3.7) n = 90	2.3 (1.6–3.7) n = 90	2.3 (1.5–3.8) n = 98	2.4 (1.5–3.9) n = 90
	*p*-value < 0.001					*p*-value < 0.001				

**Table 2 pathogens-12-00780-t002:** Baseline characteristics of patients with bloodstream infections (group, n = 246 and group, n = 154).

	Pre-Pandemic Period 2018–2019 (n = 246) Group A	COVID-19 Pandemic 2020–2022 (n = 154) Group B
Gender, male	166 (67.2%)	98 (63.6%)
Age, years, mean ± SD	65.6 (50.4–76.4)	65.8 (50.5–77.4)
Duration of bacteraemia symptoms before treatment initiation		
0–24 h	158 (64.2%)	73 (47.4%)
25–72 h	25 (10.2%)	34 (22.1%)
>72 h	55 (22.4%)	36 (23.4%)
Unknown	8 (3.2%)	11 (7.1%)
Telephone consultation	37 (15%)	117 (76%)
Bedside consultation	209 (85%)	37 (24%)

**Table 3 pathogens-12-00780-t003:** Comorbidities of patients with bloodstream infection and reported infectious disease consultation.

	Group A, 2018–2019 (n = 246)	Group B, 2020–2022 (n = 154)	*p*-Value
Operation within 30 days	34 (13.8%)	29 (18.8%)	0.04
Diabetes mellitus type 2	89 (36.2%)	68 (44.2%)	0.12
Heart failure	26 (10.6%)	19 (12.3%)	0.02
Coronary disease	49 (19.9%)	18 (11.7%)	0.45
Peripheral Vascular disease	11 (4.5%)	12 (7.8%)	0.12
Cerebrovascular disease	18 (7.3%)	17 (11%)	0.05
Chronic respiratory disease	9 (3.7%)	8 (5.2%)	0.04
Malignancies	25 (10.2%)	35 (22.7%)	0.24
Transplantation	14 (5.7%)	11 (7.14%)	1.02
Immunosuppresion	38 (15.4%)	24 (15.6%)	0.87
Chronic renal disease	22 (8.9%)	19 (12.3%)	0.04
Prosthetic device	56 (22.8%)	47 (30.5%)	0.02
Charlson comorbidity index Score ≥ 3	102 (41%)	67 (43.5%)	0.02

**Table 4 pathogens-12-00780-t004:** Clinical features of infections in adults with bloodstream infections (group, n = 246 and group, n = 154).

	Group A 2018–2019 (n = 246)	Group B 2020–2022 (n = 154)	*p*-Value
Community-acquired infection	96 (39%)	56 (36.3%)	0.001
Hospital-acquired infection	150 (61%)	98 (63.6%)	0.001
Multidrug-resistant bacteria	83 (33.7%)	57 (37%)	0.001
Focus of infection			
Unknown	16 (6.5%)	18 (11.7%)	0.004
Central venous catheter	46 (18.7%)	31 (20.1%)	0.156
Peripheral venous catheter	34 (13.8%)	21 (13.6%)	0.458
Thrombophlebitis	12 (4.9%)	27 (17.5%)	0.024
Implanted vascular device	21 (8.5%)	16 (10.4%)	0.048
Infective endocarditis	11 (4.5%)	16 (10.4%)	0.678
Native valve	6 (2.4%)	7 (4.5%)	0.465
Prosthetic valve	5 (2%)	9 (5.8%)	0.247
Joint infection	10 (4.1%)	9 (5.8%)	0.765
Prosthetic joint infection	15 (6.1%)	19 (12.3%)	0.223
Vertebral osteomyelitis	13 (5.3%)	17 (11%)	0.058
Intra-abdominal infections	26 (10%)	18 (11.7%)	0.047
Osteomyelitis/diabetic foot ulcers	29 (11.8%)	20 (13%)	0.023
Skin and soft-tissue infections	24 (9.8%)	19 (12.3%)	0.027
Respiratory infections	32 (13%)	21 (13.6%)	0.057
Urinary tract infections	19 (7.7%)	16 (10.4%)	0.077
Central nervous system infections	9 (3.7%)	7 (4.5%)	0.065
Complicated infection	134 (54.5%)	87 (56.5%)	0.001

**Table 5 pathogens-12-00780-t005:** Clinical indicators and outcomes in the management of bloodstream infections (group, n = 246 and group, n = 154).

	Group A 2018–2019 (n = 246)	Group B 2020–2022 (n = 154)	*p*-Value
Septic shock	8 (3.3%)	7 (4.5%)	0.118
Hospitalization in ICU	11 (4.5%)	12 (7.8%)	0.245
Hospital stay, days, mean ± SD	29 (17–52)	30 (16–51)	0.457
Mortality			
Within 28 days	12 (4.9%)	16 (10.4%)	0.001
Within 90 days	19 (7.7%)	23 (14.9%)	0.001
Repeated blood culture	137 (55.7%)	56 (36.4%)	0.001
Negative blood culture within 7 days	98 (40%)	48 (31.2%)	0.001
Recurrent disease	9 (3.6%)	6 (3.9%)	0.458
Duration of antibiotic treatment, days, mean ± SD	15 (8–19)	11 (6–12)	0.04
Repeated clinical estimation	112 (45.5%)	36 (23.4%)	0.001
Combination of antibiotics	26 (10.6%)	11 (7.1%)	0.001
Recorded bloodstream infection (isolated pathogen) in discharge summary	124 (50.4%)	44 (28.6%)	0.001

## Data Availability

Not applicable.

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
