# Peer review of "The Impact of the COVID-19 Pandemic on Antimicrobial Resistance and Management of Bloodstream Infections"

_pathogens, 2023, doi:10.3390/pathogens12060780_

Round 1

Reviewer 1 Report

The manuscript presents a descriptive study that aims to investigate the impact of the COVID-19 pandemic on the burden of antimicrobial resistance in hospitalized patients. While the topic is important, the study's design contains some major flaws.

Major Comments

Firstly, the authors compared the percentage of non-susceptible isolates in several gram-negative species recovered from blood culture and respiratory culture before and during the pandemic to demonstrate a significant increase in the rates of multi-drug resistant bacteria during COVID-19. However, it's important to note that the study was not designed as a controlled study. Uncontrolled variables such as patient population, length of hospital stay, and infection duration may have contributed to the increase in the number of resistant isolates. For instance, the significant increase in the number of meropenem-resistant K. pneumoniae during the pandemic may be due to the fact that more patients from long-term care facilities were hospitalized during the pandemic. Therefore, simply comparing the rate of multi-drug resistant isolates is insufficient to support the author's conclusion.

Secondly, the manuscript lacks definitions for key terms such as community-acquired infection and hospital-acquired infection. Furthermore, there is no clear criteria provided for obtaining infectious disease consultations. The lack of access to infectious disease consults, rather than telephone consultations, may contribute to poor outcomes among patients with bloodstream infections.

Reviewer 2 Report

This manuscript from Greece documents significantly more infections by gram-negative isolates during the COVID-19 pandemic compared to prior to the pandemic.  A few comments for the authors to consider:

1.  How does the changes in the susceptibility profiles of the gram-negative isolates impact these results.  The European Committee for Standardization made changes to the susceptibility profiles of some of these organisms.  How does this impact these results?  Authors need to comment. 

2.  some of the isolate numbers particularly in the ICU are low and thus may not represent true resistance.  How does this impact these results?  Authors need to comment. 

3.  The authors need to provide some demographics of their health system - how many total beds, how many ICU beds, occupancy of the hospital during the 2 data collection periods. 

4.  May need to changes some of the tables to Western characters (p-values). 

Item #4 is more for the editors but authors need to know also. 

Reviewer 3 Report

Quite impressive work to compile available data from bacteriology investigations in one hospital in Greece and with worrying trend of increasing degrees of multi drug resistant bacteria of high concern. Some points to consider however:

1. In this report there is a definitely clear tendency that resistance to most antibiotics increase during the pandemic period. This is in contrast with the even larger survey of hospitals in Greece (ref 1), where resistance frequencies actually decreased in some cases. This could be pointed out better in the discussion, since it may point to something specific for the hospital studied here.

2. One strong point of this publication compared to ref 1 is the inclusion of several other factors that may be affected by the disturbance caused by the pandemic. However, it is somewhat difficult to extract exactly what the authors consider important information and again the discussion should bring this out better. Alternatively, the amount of information in the tables could be limited to the most relevant information.

3. Tables 1 and 2 could be better presented as figures, will show the trends more clearly.

Language should be checked by a native English speaking person. The text is understandable, but language use could be improved.

Reviewer 4 Report

General comments

This retrospective longitudinal study analysed bloodstream infections and ID expert consultancies with regards to these infections before and during the pandemic in Greece.  Some definitions are missing (line 101-102), like what threshold (days after admission, …) and parameters (invasive device present, central line, secondary infections, …) were applied to define a healthcare/hospital acquired infection. The statistical analysis is also strange presented, in particular the univariate analysis and multivariate model and results shown.

For these and the comments in detail below, the article requires major revisions before a reconsideration for publication can be done.

Detailed comments

The English language should be carefully checked throughout the document.

Abstract: line 13-15 is quite similar to your current expressed conclusion of the study here presented and thus rephrase the background.

Introduction

Line 42-43- inappropriate reference for the first sentence, please modify .

Line 45: check if used reference for multidrug-resistance during the pandemic (WHONet GREECE) also  have studied mortality, morbidity and economic consequences and consider to add for this additional references.

Line 47: include reference year of this estimated mortality.

Line 54-58: an ECDC update has been published in Nov 2022, consider to modify accordingly.

Line 59-60: provide appropriate reference that studied this causal relationship or otherwise rephrase as a potential/plausible relationship - correlation, or move to the discussion.

Line 64: check that all Latin bacterial names are spelled italic, throughout the manuscript, figures and tables (e.g. 89-91)

Line 71-72:  specify what you want to measure related to antimicrobial resistance (incidence?)

Line 74-79: consider to move this to the discussion (alternatively move up stressing the situation in Greece when the global situation is introduced). Include appropriated references for the statements made in this paragraph.

Line 88: plural form for ICU might be more appropriate (intensive care units, ICUs)

Line 102-103: include what definition of healthcare-associated infection was applied, and provide appropriate reference. (only HA-BSI?, CLABSI?, other?) -  also see line 111.

Line 113: if applicable, specify or be more concrete: ‘product, formulation, route of administration, dose, duration, treatment interval,…? Als what unit was applied to analyse this (binomial, treatment days, DDD, ???)

Line 124-128: include for period A and B, the initial and cleaned number of patients involved (the number of duplicates removed during descriptive analysis).

Line 134 & 163…: ‘was’ can be removed

Figures & Tables: include in all titles shortly the objective, the period and region and the number of patients involved. Latin names should be written italic. Also abbreviations (e.g. ICU) should be repeated for each individual table/legend.

The p-values in table 5-7 look awkward and should be carefully rechecked. The multivariate model does not take into account comorbidities and antimicrobial resistance, while the conclusions are based upon this assumption. This should be carefully reconsidered.

Figures: add the clearly the applied definition of ‘wards’ in the legend

see above

Round 2

Reviewer 1 Report

The concerns raised by the reviewer have been adequately and effectively addressed.

Author Response

On behalf of authors, we would like to thank the reviewer for the comments.

Yours sincerely,

Petrakis Vasileios

Reviewer 4 Report

Dear author, 

The review was done too fast and with insufficient detail as requested. Eg. the multivariate analysis, which is insufficiently outlined (forward/backward, what variables, number of patients,... ) in both the document and the corresponding table. As such, the analysis should be removed (Table 4) or clarified. Please apply a color code for the figures. 

Kind regards 
